# CT-derived body composition: Differential association with disease, age and inflammation in a retrospective cohort study

**Nicholas A. Bradley** [1]*, **Josh McGovern**[1], **Ross D. Dolan**[1], **Allan M. Golder**[1], **Campbell S. D. Roxburgh**[1], **Graeme J. K. Guthrie**[2], **Donald C. McMillan**[1]

**1** Academic Unit of Surgery, University of Glasgow, Glasgow, United Kingdom, **2** Department of Vascular Surgery, NHS Tayside, Dundee, United Kingdom

* Nicholasandrew.bradley@glasgow.ac.uk

**Data Availability Statement:** The data used to generate this work contain sensitive clinical information for patients undergoing procedures as

## Abstract

### Background

Low skeletal muscle mass and density, as assessed by CT-body composition (CT-BC), are recognised to have prognostic value in non-cancer and cancer patients. The aim of the present study was to compare CT-BC parameters between non-cancer (abdominal aortic aneurysm, AAA) and cancer (colorectal cancer, CRC) patients.

### Methods

Two retrospective multicentre cohorts were compared. Thresholds of visceral fat area (VFA, Doyle), skeletal fat index (SFI, Ebadi), skeletal muscle index (SMI, Martin), and skeletal muscle density (SMD, Martin) were applied to these cohorts and compared. The systemic inflammatory response (SIR) was measured by the systemic inflammatory grade (SIG).

### Results

1695 patients were included; 759 patients with AAA and 936 patients with CRC. Low SMD (33% vs. 66%, $p$ <0.001) was more prevalent in the CRC cohort. Low SMI prevalence was similar in both cohorts (51% vs. 51%, $p$ = 0.80). Compared with the AAA cohort, the CRC cohort had a higher prevalence of raised SIG ($p$ <0.001). Increasing age (OR 1.54, 95% CI 1.38–1.72, $p$ < 0.001) and elevated SIG (OR 1.23, 95% CI 1.09–1.40, $p$ = 0.001) were independently associated with increased odds of low SMI. Increasing age (OR 1.90, 95% CI 1.66–2.17, $p$ < 0.001) CRC diagnosis (OR 5.89, 95% CI 4.55–7.62, $p$ < 0.001), ASA > 2 (OR 1.37, 95% CI 1.08–1.73, $p$ = 0.01), and elevated SIG (OR 1.19, 95% CI 1.03–1.37, $p$ = 0.02) were independently associated with increased odds of low SMD.

### Conclusions

Increasing age and systemic inflammation appear to be important determinants of loss of skeletal muscle mass and quality irrespective of disease.

documented in the manuscript. These data are also being used to support submission of a PhD thesis at the University of Glasgow. In line with the policy of the ethics committee who approved the study (see above), and university data governance policies, in order to share data a formal data sharing agreement is required. Request for data sharing can be made either to the corresponding author who can implement request for data sharing agreement, or alternatively to the University of Glasgow MVLS contracts team (see https://www.gla.ac.uk/myglasgow/researchsupportoffice/contracts-team/#collegeofmvlsenquiries for contact details). The data will be stored in the University of Glasgow Enlighten data repository (https://www.gla.ac.uk/myglasgow/openresearch/researchdatamanagement/enlightenresearchdatainformation/).

**Funding:** The author(s) received no specific funding for this work.

**Competing interests:** The authors have declared that no competing interests exist.

## Introduction

Abdominal aortic aneurysm (AAA) is a condition with a UK prevalence of 1.5%, rising to 2–4% in males aged over 65 [1]. Risk of rupture increases with increased diameter, and current typical UK practice is to consider intervention in AAA over 5.5cm, with the choice of endovascular aneurysm repair (EVAR) or open surgical repair (OSR) based on aneurysm morphology, patient fitness, and comorbidity [2]. Colorectal cancer (CRC) is the fourth most commonly diagnosed cancer worldwide, with an increased incidence in males and in older adults [3]. CRC is staged using the AJCC TNM (8th Edition) staging system [4], with curative surgical resection the gold standard of treatment for early stage resectable disease [5]. Both AAA and CRC are considered to occur on a background of older age, obesity, comorbid disease and functional decline [6, 7].

Sarcopenia is defined as a progressive loss of skeletal muscle volume and progressive reduction in skeletal muscle function [8], and can be assessed radiologically through CT body composition analysis (CT-BC), which allows for quantification of sarcopenia using the skeletal muscle index (SMI) and the skeletal muscle density (SMD), typically at the L3 vertebral level. Low skeletal muscle mass, as measured by SMI, has been found to be prevalent in both cancer and non-cancer patient groups, and appears to confer a worse prognosis in both of these cohorts [9, 10]. Low SMD appears to be associated with increased lipid content within myocytes [11], and may therefore represent poor quality muscle through accumulation of fat droplets in otherwise healthy muscle tissue. Low SMD is also associated with inferior prognosis in cancer and non-cancer populations [9, 12].

Chronic activation of the systemic inflammatory response (SIR) is evident in both cardiovascular disease and cancer [13]. In atherosclerotic disease, chronic inflammation is increasingly being recognised as the major factor driving disease progression [14]. In both patients with cardiovascular disease and cancer, chronic inflammation confers inferior prognosis [15, 16], and an association between systemic inflammation and altered CT-BC has been described [17, 18].

The prevalence of altered CT-BC parameters in a range of disease states questions whether body composition is primarily constitutional and related to shared risk factors, or whether it results from disease specific factors. For example, a recent systematic review has suggested that CT-derived sarcopenia has a similar prevalence across a range of different cancers and disease stages [19]. There have been attempts to define normal values of CT-BC parameters using "healthy" control populations [20, 21], however when comparing these cohorts to clinically relevant disease entities such as CRC, there are likely to be potentially confounding factors.

The aim of the present study was to compare CT-derived body composition and systemic inflammation in two large cohorts of patients undergoing elective intervention for AAA and elective curative resection for colorectal cancer.

## Patients & methods

### Patients

The present study included 2 large multicentre cohorts of patients. The AAA cohort was retrospectively recruited from 3 large tertiary vascular units in Scotland using existing institutional operative logbooks; consecutive patients undergoing EVAR, F/B-EVAR, or OSR for AAA between 01/01/2015 and 10/01/2021 were screened for eligibility. Emergency cases were excluded due to the confounding effect on other covariates. West of Scotland Research Ethics Committee approval was obtained for this patient cohort (Reference 21/WS/0146; approval granted 23/11/2021).

The CRC cohort was recruited from prospectively recorded cancer network data; within Scotland, basic clinicopathological data for all new cases of colorectal cancer are prospectively collected and held within regional Managed Clinical Network (MCN) datasets. The West of Scotland Cancer Network (WoSCAN) dataset incorporates four health boards (Ayrshire and Arran, Forth Valley, Lanarkshire and Greater Glasgow and Clyde) and includes approximately half of the population of Scotland. These patients receive treatment in line with national guidelines and are followed up for a period of 3–5 years. Patients diagnosed with colon cancer between January 2011 and December 2014 were identified from the West of Scotland MCN database. Patients undergoing curative elective surgery for a diagnosis of AJCC stage I–III [4] colon cancer were included. Ethical approval was granted for this project from the Public Benefit and Privacy Panel (PBPP) for Health and Social Care (NHS Scotland) and Caldicott Guardian Approval.

## Data collection

Baseline demographic and clinicopathological data were collected from hospital medical notes and community health records. Comorbidity was assessed using ASA derived from preoperative anaesthetic assessment. Age ($< 60$, 60–69, 70–79, $\geq 80$), ASA ($\leq 2$ and $> 2$), and BMI ($<25$ and $\geq 25 kg/m^2$) were considered as categorical variables. Haemoglobin and creatinine were recorded from routine pre-operative blood tests and defined as "normal" or "high/low" based on local laboratory sex-specific reference values. Survival data were obtained from the Community Health Index (CHI) registry, a routinely available registry maintained at a national health board level and populated from both primary and secondary care data. Specific cause of death was not available from this registry. Confidential data were accessed on 03/06/2023, and authors had access to identifiable data during but not after data collection.

## SIR

Routine pre-operative blood tests performed as part of existing patient care were used to assess the SIR; the modified Glasgow Prognostic Score (mGPS, based on CRP and albumin) [15] and neutrophil:lymphocyte ratio (NLR, based on differential white cell count) [22] were calculated as previously described and combined as described by Golder *et al* into the systemic inflammatory grade (SIG, S1 Table) [23]. Patients were sub-grouped based on SIG into SIG 0 (non-inflamed), SIG 1 (mildly inflamed) and SIG $\geq 2$ (inflamed).

## CT-BC

CT-BC was performed using established methodology as previously reported [24]. Briefly, the free to access Java-based program ImageJ v1.47 was used to manually segment muscle and adipose tissue on pre-operative CTs performed as part of existing patient care, within 3 months prior to intervention. The subcutaneous fat index (SFI) and SMI were recorded based on raw measurements normalised to patient height$^2$ ($m^2$), whilst the visceral fat area (VFA) ($cm^2$) and SMD (HU) were not normalised in keeping with previous literature. CT-BC parameters were then sub-grouped based on previously reported thresholds [25–27] (S2 Table). Compromised images where image quality precluded accurate CT-BC were excluded. Analysis was performed on arterial phase scans (AAA cohort) and portal venous phase scans (CRC) cohort; in order to mitigate the potential risk of bias in SMD readings between these 2 phases SMD analyses were repeated using values in the AAA cohort transformed from arterial phase to comparable portal venous phase as per Rollins *et al* (SMD$_{Rollins}$) [28].

## Statistical analyses

Analyses were performed on the entire study cohort, and subsequently adjusted analyses were performed on patients aged 70–79 years (chosen as the sub-group of age with maximal patient numbers), to control for potential confounding due to differences in age distribution between the AAA and CRC cohorts. Differences between categorical variables were assessed using the Chi-Squared Test, with 2-tailed linear-by-linear association $p$ values reported. Differences between continuous variables were assessed using the Mann-Whitney U Test, with median (interquartile range, IQR) reported. The predictive value of demographic and clinicopathological characteristics on CT-BC parameters was assessed using binary logistic regression. AAA repair strategy (EVAR/OSR) and AJCC stage were not included in multivariate models to avoid excluding the entire sub-group of CRC or AAA patients respectively. Binary logistic regression results are displayed as odds ratio (OR), 95% CI, and $p$ value. Missing parameters were selectively excluded from relevant analyses on a case-by-case basis. $p < 0.05$ was considered statistically significant. Statistical analyses were conducted using IBM SPSS v28.0.

## Results

Following exclusions there were a total of 1695 patients who underwent elective intervention for either AAA or CRC included; 759 patients with AAA and 936 patients with CRC. In the AAA cohort, 604 (80%) underwent EVAR compared with 155 (20%) undergoing OSR. In the CRC cohort, 224 (24%) were AJCC stage I, 380 (41%) were AJCC stage II, and 332 (35%) were AJCC stage III. When the cohort was adjusted for age, there were 672 patients included in age 70–79 sub-group analysis; 359 patients with AAA and 313 patients with CRC (S1 Fig).

The difference in baseline demographic and clinicopathological characteristics, CT-BC parameters, markers of the SIR, and survival between AAA and CRC in the entire study cohort is shown in Table 1. Compared with the AAA cohort, patients with CRC were younger ($p < 0.001$), more likely to be female ($p < 0.001$), had a lower ASA ($p < 0.001$), had a lower BMI ($p < 0.001$), had a lower haemoglobin ($p < 0.001$), and were less likely to have a raised creatinine ($p < 0.001$). High VFA (79% vs. 73%, $p = 0.006$) was more prevalent in the AAA cohort, whilst high SFI (74% vs. 81%, $p < 0.001$) and low SMD (33% vs. 66%, $p < 0.001$) were more prevalent in the CRC cohort. Low SMI prevalence was similar in both cohorts (51% vs. 51%, $p = 0.80$). Compared with the AAA cohort, the CRC cohort had a higher prevalence of raised SIG ($p < 0.001$). 30 day and 1-year outcomes were similar between groups, however the AAA cohort had decreased 3-year survival (76% vs. 83%, $p = 0.001$).

The difference in baseline demographic and clinicopathological characteristics, markers of the SIR, and survival between AAA and CRC in the age 70–79 sub-group is shown in Table 2. Following adjustment, median age was similar ($p = 0.55$) between patients with AAA and CRC. Compared with the AAA cohort, patients with CRC were more likely to be female ($p < 0.001$), and had a lower haemoglobin ($p < 0.001$). Compared with the AAA cohort, the CRC cohort had a higher prevalence of raised SIG ($p < 0.001$). 1-year and 3-year outcomes were similar between groups; however the AAA cohort had decreased 30-day survival (98% vs. 100%, $p = 0.01$).

The difference in BMI and CT-BC parameters between AAA and CRC in the entire study cohort sub-grouped by age is shown in Table 3. Low SMD was more prevalent in the CRC cohort across all age sub-groups. BMI was significantly higher in patients with AAA apart from those aged <60, where a non-significant trend was observed ($p = 0.07$). High SFI was more prevalent in the CRC sub-group in patients aged 60–69 ($p = 0.03$), however was similar between patients with AAA and CRC of other age groups.

**Table 1.** The difference between demographic and clinicopathological characteristics, CT-BC and systemic inflammation in patients undergoing elective EVAR or OSR for AAA or curative resection for CRC (n = 1695).

| | AAA | Curative CRC | *p* |
|---|---|---|---|
| | (n = 759) | (n = 936) | |
| **Age** | | | |
| < 60 | 12 (2%) | 207 (22%) | <0.001 |
| 60–69 | 222 (29%) | 295 (32%) | |
| 70–79 | 359 (47%) | 313 (33%) | |
| ≥ 80 | 166 (22%) | 121 (13%) | |
| **Sex** | | | |
| Male | 703 (93%) | 520 (55%) | <0.001 |
| Female | 56 (7%) | 416 (45%) | |
| **ASA** | | | |
| ≤ 2 | 418 (55%) | 614 (66%) | <0.001 |
| > 2 | 337 (45%) | 322 (34%) | |
| **BMI** | | | |
| < 25 kg/m$^2$ | 158 (21%) | 320 (34%) | <0.001 |
| ≥ 25 kg/m$^2$ | 596 (79%) | 616 (66%) | |
| **AJCC Stage** | | | |
| I | - | 224 (24%) | - |
| II | - | 380 (41%) | |
| III | - | 332 (35%) | |
| **Repair Strategy** | | | |
| EVAR | 604 (80%) | - | - |
| OSR | 155 (20%) | - | |
| **Haemoglobin** | | | |
| Normal | 606 (81%) | 515 (61%) | <0.001 |
| Low | 146 (19%) | 327 (39%) | |
| **Creatinine** | | | |
| Normal | 588 (78%) | 724 (88%) | <0.001 |
| High | 165 (22%) | 103 (12%) | |
| **SFI (Ebadi)** | | | |
| Normal | 184 (26%) | 180 (19%) | <0.001 |
| High | 522 (74%) | 756 (81%) | |
| **VFA (Doyle)** | | | |
| Normal | 157 (21%) | 250 (27%) | **0.006** |
| High | 594 (79%) | 686 (73%) | |
| **SMI (Martin)** | | | |
| Normal | 357 (49%) | 461 (49%) | 0.80 |
| Low | 377 (51%) | 475 (51%) | |
| **SMD (Martin)** | | | |
| Normal | 495 (67%) | 316 (34%) | <0.001 |
| Low | 244 (33%) | 620 (66%) | |
| **SIG** | | | |
| 0 | 331 (52%) | 408 (44%) | <0.001 |
| 1 | 207 (33%) | 236 (25%) | |
| ≥ 2 | 93 (15%) | 286 (31%) | |
| **30-Day Mortality** | | | |

(*Continued*)

**Table 1.** (Continued)

| | AAA | Curative CRC | p |
|---|---|---|---|
| | (n = 759) | (n = 936) | |
| Yes | 10 (1%) | 6 (1%) | 0.15 |
| No | 749 (99%) | 930 (99%) | |
| **1-Year Survival** | | | |
| Yes | 717 (95%) | 881 (94%) | 0.76 |
| No | 43 (5%) | 55 (6%) | |
| **3-Year Survival** | | | |
| Yes | 580 (76%) | 775 (83%) | **0.001** |
| No | 179 (24%) | 161 (17%) | |

p values generated through Chi-Squared analyses comparing proportions of each variable between the AAA and CRC cohorts.

AAA: abdominal aortic aneurysm. CRC: colorectal cancer. BMI: body mass index. SIG: systemic inflammatory grade. SFI: subcutaneous fat index. VFA: visceral fat area. SMI: skeletal muscle index. SMD: skeletal muscle density. EVAR: endovascular aneurysm repair. OSR: open surgical repair. AJCC: American Joint Committee on Cancer.

The binary logistic regression models for the association between baseline demographic and clinicopathological characteristics, systemic inflammation, and CT-BC muscle parameters is shown in Table 4. On univariate analysis, increasing age ($p < 0.001$), AJCC Stage II/III ($p = 0.02$), and elevated SIG ($p < 0.001$) were associated with increased odds of low SMI, whilst BMI $\geq$ 25 kg/m$^2$ ($p = 0.01$) was associated with decreased odds of low SMI. On multivariate analysis, increasing age (OR 1.54, 95% CI 1.38–1.72, $p < 0.001$) and elevated SIG (OR 1.23, 95% CI 1.09–1.40, $p = 0.001$) were independently associated with increased odds of low SMI, whilst BMI $\geq$ 25 kg/m$^2$ (OR 0.78, 95% CI 0.62–0.98, $p = 0.03$) was independently associated with decreased odds of low SMI. On univariate analysis, increasing age ($p < 0.001$), CRC diagnosis ($p < 0.001$), ASA > 2 ($p = 0.007$), and elevated SIG ($p < 0.001$) were associated with increased odds of low SMD, whilst BMI $\geq$ 25 kg/m$^2$ ($p < 0.05$) and OSR ($p < 0.001$) were associated with decreased odds of low SMD. On multivariate analysis, increasing age (OR 1.90, 95% CI 1.66–2.17, $p < 0.001$) CRC diagnosis (OR 5.89, 95% CI 4.55–7.62, $p < 0.001$), ASA > 2 (OR 1.37, 95% CI 1.08–1.73, $p = 0.01$), and elevated SIG (OR 1.19, 95% CI 1.03–1.37, $p = 0.02$) were independently associated with increased odds of low SMD, whilst BMI $\geq$ 25 kg/m$^2$ (OR 0.51, 95% CI 0.39–0.66, $p < 0.001$) was independently associated with decreased odds of low SMD.

Fig 1 displays the proportion of patients with abnormal BMI and CT-BC parameters subgrouped by age (Fig 1a) and SIG (Fig 1b) in the entire study cohort. The prevalence of low SMI and SMD was associated with increasing age in both the AAA and CRC cohorts, with a greater magnitude of association between age and SMD in patients with CRC compared to AAA. As SIG increased, there was a trend towards a higher proportion of patients with low SMD in both AAA and CRC cohorts. Compared with the AAA cohort, the proportion of patients with low SMD in the CRC cohort was consistently higher across each SIG subgroup.

Fig 2 displays the proportion of patients with abnormal BMI and CT-BC parameters subgrouped by SIG in the age 70–79 sub-group. As SIG increased, there was a trend towards a higher proportion of patients with low SMI in the CRC cohort, and a low SMD in both AAA and CRC cohorts. Compared with the AAA cohort, the proportion of patients with low SMD in the CRC cohort was consistently higher across each SIG subgroup.

**Table 2. The difference between demographic and clinicopathological characteristics and systemic inflammation in patients aged 70–79 undergoing elective EVAR or OSR for AAA or curative resection for CRC (n = 672).**

|  | AAA | Curative CRC | $p$ |
|---|---|---|---|
|  | (n = 359) | (n = 313) |  |
| **Median (IQR) Age (years)** | 74 (5) | 74 (5) | 0.55 |
| **Sex** |  |  |  |
| Male | 330 (92%) | 173 (55%) | **<0.001** |
| Female | 29 (8%) | 140 (45%) |  |
| **ASA** |  |  |  |
| $\leq 2$ | 194 (54%) | 176 (56%) | 0.60 |
| $> 2$ | 164 (46%) | 137 (44%) |  |
| **AJCC Stage** |  |  |  |
| I | - | 76 (24%) | - |
| II | - | 124 (40%) |  |
| III | - | 113 (36%) |  |
| **Repair Strategy** |  |  |  |
| EVAR | 292 (81%) | - | - |
| OSR | 67 (19%) | - |  |
| **Haemoglobin** |  |  |  |
| Normal | 276 (78%) | 159 (57%) | **<0.001** |
| Low | 80 (22%) | 118 (43%) |  |
| **Creatinine** |  |  |  |
| Normal | 281 (79%) | 230 (83%) | 0.23 |
| High | 75 (21%) | 48 (17%) |  |
| **SIG** |  |  |  |
| 0 | 147 (51%) | 125 (40%) | **<0.001** |
| 1 | 102 (35%) | 85 (27%) |  |
| $\geq 2$ | 39 (14%) | 103 (33%) |  |
| **30-Day Mortality** |  |  |  |
| Yes | 7 (2%) | 0 (0%) | **0.01** |
| No | 352 (98%) | 313 (100%) |  |
| **1-Year Survival** |  |  | 0.44 |
| Yes | 339 (94%) | 291 (93%) |  |
| No | 20 (6%) | 22 (7%) |  |
| **3-Year Survival** |  |  | 0.44 |
| Yes | 279 (78%) | 251 (80%) |  |
| No | 80 (22%) | 62 (20%) |  |

$p$ values generated through Chi-Squared analyses comparing proportions of each variable between the AAA and CRC cohorts (categorical variables), and Mann-Whitney Test (continuous variables).

AAA: abdominal aortic aneurysm. CRC: colorectal cancer. BMI: body mass index. SIG: systemic inflammatory grade. SFI: subcutaneous fat index. VFA: visceral fat area. SMI: skeletal muscle index. SMD: skeletal muscle density. EVAR: endovascular aneurysm repair. OSR: open surgical repair. AJCC: American Joint Committee on Cancer.

When the SMD analyses were repeated using $SMD_{Rollins}$, low SMD was less prevalent in the AAA cohort (21% vs. 66%, $p < 0.001$). To account for potential categorical misclassification based on phase of scan, continuous values of SMD and $SMD_{Rollins}$ were analysed. Median (IQR) SMD in the AAA vs. CRC cohorts was 37.8 (9.5) HU vs. 31.8 (12.7) HU ($p < 0.001$). Median (IQR) $SMD_{Rollins}$ in the AAA vs. CRC cohorts was 40.2 (9.3) HU vs. 31.8 (12.7) HU ($p < 0.001$).

**Table 3. The difference between BMI and CT-BC parameters in patients undergoing elective EVAR or OSR for AAA or curative resection for CR, sub-grouped by age (n = 1695).**

| | AAA | Curative CRC | p |
|---|---|---|---|
| **< 60 Years** | n = 12 | n = 207 | |
| **BMI** | | | |
| < 25 kg/m$^2$ | 1 (8%) | 69 (33%) | 0.07 |
| ≥ 25 kg/m$^2$ | 11 (92%) | 138 (67%) | |
| **SFI (Ebadi)** | | | |
| Normal | 1 (8%) | 48 (23%) | 0.23 |
| High | 11 (92%) | 159 (77%) | |
| **VFA (Doyle)** | | | |
| Normal | 2 (17%) | 73 (35%) | 0.19 |
| High | 10 (83%) | 134 (65%) | |
| **SMI (Martin)** | | | |
| Normal | 10 (83%) | 132 (64%) | 0.17 |
| Low | 2 (17%) | 75 (36%) | |
| **SMD (Martin)** | | | |
| Normal | 11 (92%) | 118 (57%) | **0.02** |
| Low | 1 (8%) | 89 (43%) | |
| **60–69 Years** | n = 222 | n = 295 | |
| **BMI** | | | |
| < 25 kg/m$^2$ | 33 (15%) | 85 (29%) | **<0.001** |
| ≥ 25 kg/m$^2$ | 187 (85%) | 210 (71%) | |
| **SFI (Ebadi)** | | | |
| Normal | 44 (22%) | 42 (14%) | **0.03** |
| High | 158 (78%) | 253 (86%) | |
| **VFA (Doyle)** | 39 (18%) | 66 (22%) | 0.20 |
| Normal | 181 (82%) | 229 (78%) | |
| High | | | |
| **SMI (Martin)** | 127 (59%) | 158 (54%) | 0.20 |
| Normal | 87 (41%) | 137 (46%) | |
| Low | | | |
| **SMD (Martin)** | | | **<0.001** |
| Normal | 163 (76%) | 115 (39%) | |
| Low | 52 (24%) | 180 (61%) | |
| **70–79 Years** | n = 359 | n = 313 | |
| **BMI** | | | **<0.001** |
| < 25 kg/m$^2$ | 70 (20%) | 108 (35%) | |
| ≥ 25 kg/m$^2$ | 287 (80%) | 205 (65%) | |
| **SFI (Ebadi)** | | | |
| Normal | 79 (24%) | 57 (18%) | 0.09 |
| High | 255 (76%) | 256 (82%) | |
| **VFA (Doyle)** | | | 0.53 |
| Normal | 74 (21%) | 72 (23%) | |
| High | 279 (79%) | 241 (77%) | |
| **SMI (Martin)** | | | |
| Normal | 162 (47%) | 134 (43%) | 0.30 |
| Low | 184 (53%) | 179 (57%) | |
| **SMD (Martin)** | | | |

*(Continued)*

**Table 3.** (Continued)

| | AAA | Curative CRC | *p* | |
|---|---|---|---|---|
| Normal | 237 (68%) | 72 (23%) | <**0.001** | |
| Low | 112 (32%) | 241 (77%) | | |
| ≥ 80 Years | n = 166 | n = 121 | | |
| **BMI** | | | | |
| < 25 kg/m$^2$ | 54 (33%) | 58 (48%) | <**0.001** | |
| ≥ 25 kg/m$^2$ | 111 (67%) | 63 (52%) | | |
| **SFI (Ebadi)** | | | | |
| Normal | 60 (38%) | 33 (27%) | 0.06 | |
| High | 98 (62%) | 88 (73%) | | |
| **VFA (Doyle)** | | | | |
| Normal | 42 (25%) | 39 (32%) | 0.20 | |
| High | 124 (75%) | 82 (68%) | | |
| **SMI (Martin)** | | | | |
| Normal | 58 (36%) | 37 (31%) | 0.36 | |
| Low | 104 (64%) | 84 (69%) | | |
| **SMD (Martin)** | | | | |
| Normal | 84 (52%) | 11 (9%) | <**0.001** | |
| Low | 79 (48%) | 110 (91%) | | |

*p* values generated through Chi-Squared analyses comparing proportions of each variable between the AAA and CRC cohorts (categorical variables). SFI: subcutaneous fat index. VFA: visceral fat area. SMI: skeletal muscle index. SMD: skeletal muscle density.

## Discussion

To our knowledge this is the first study to directly compare baseline demographic and clinico-pathological characteristics CT-BC parameters and systemic inflammation between large cohorts of patients with AAA and CRC. The results of the present study describe similar rates of low SMI in the AAA and CRC cohorts. In contrast, low SMD and elevated magnitude of the SIR were more prevalent in patients with CRC compared with AAA, despite the greater BMI in the AAA cohort. An association between age and both low SMI and low SMD was also observed, in both patients with AAA and CRC. Therefore, loss of skeletal muscle mass appears to be endemic across multiple chronic illnesses, however loss of skeletal muscle density exhibits greater disease-specific variation. Moreover, it would appear that increasing age and systemic inflammation predominates in determining loss of skeletal muscle mass and quality.

The present results support the "common soil" hypothesis of shared aetiology and risk factors producing similar host phenotypes despite different disease entities [29]. These results are consistent to those in patients with cancer, McGovern *et al* [19] reported similar rates of low skeletal muscle mass in different cancer subtypes and disease stages. Therefore, taken together these results would suggest that sarcopenia is endemic across a range of common, life-limiting diseases, and highlight age and systemic inflammation as key determinants of altered body composition and a host response rather than disease-specific entity. The disease-specific differences in SMD observed in the present study may be in part explained by tumour mediated inflammation, however these differences were independent of inflammatory status. It may be that the inflammatory cascade in patients with cancer include alternative pathways which are not entirely captured through SIG. Further prospective analysis, including cytokine analysis, is required to more accurately define these pathways.

**Table 4. The association between baseline demographic and clinicopathological covariates, systemic inflammation, and CT-BC parameters in patients undergoing elective intervention for AAA or curative resection for CRC (n = 1695).**

| | N (%) | Low SMI (Martin) | | Low SMD (Martin) | |
|---|---|---|---|---|---|
| | | Univariate | Multivariate | Univariate | Multivariate |
| **Age**<br>(< 60, 60–69, 70–79, ≥ 80) | 219 (13%) / 517 (30%) / 672 (40%) / 287 (17%) | 1.55<br>1.39–1.73<br>$p < 0.001$ | 1.54<br>1.38–1.72<br>$p < 0.001$ | 1.41<br>1.27–1.57<br>$p < 0.001$ | 1.90<br>1.66–2.17<br>$p < 0.001$ |
| **CRC**<br>(Yes / No) | 936 (55%) / 759 (45%) | 0.98<br>0.80–1.18<br>$p = 0.80$ | - | 3.98<br>3.24–4.88<br>$p < 0.001$ | 5.89<br>4.55–7.62<br>$p < 0.001$ |
| **ASA**<br>(≤ 2 / > 2) | 1032 (61%) / 659 (39%) | 1.07<br>0.88–1.30<br>$p = 0.52$ | - | 1.31<br>1.08–1.60<br>$p = 0.007$ | 1.37<br>1.08–1.73<br>$p = 0.01$ |
| **BMI**<br>(<25 kg/m² / ≥ 25 kg/m²) | 478 (28%) / 1212 (72%) | 0.76<br>0.61–0.94<br>$p = 0.01$ | 0.78<br>0.62–0.98<br>$p = 0.03$ | 0.40<br>0.32–0.50<br>$p < 0.001$ | 0.51<br>0.39–0.66<br>$p < 0.001$ |
| **OSR**[†]<br>(Yes / No) | 155 (20%) / 604 (80%) | 0.87<br>0.61–1.25<br>$p = 0.87$ | - | 0.45<br>0.29–0.69<br>$p < 0.001$ | - |
| **AJCC Stage**[†]<br>(I / II & III) | 224 (24%) / 712 (76%) | 1.45<br>1.07–1.96<br>$p = 0.02$ | - | 1.27<br>0.93–1.74<br>$p = 0.13$ | - |
| **SIG**<br>(0 / 1 / ≥ 2) | 739 (47%) / 443 (28%) / 379 (25%) | 1.27<br>1.13–1.44<br>$p < 0.001$ | 1.23<br>1.09–1.40<br>$p = 0.001$ | 1.43<br>1.26–1.62<br>$p < 0.001$ | 1.19<br>1.03–1.37<br>$p = 0.02$ |

Data presented are Odds Ratio (OR), 95% CI, *p* value. For covariates with >2 subgroups, the first category was considered as the reference category. CRC: colorectal cancer. BMI: body mass index. SIG: systemic inflammatory grade. SMI: skeletal muscle index. SMD: skeletal muscle density.

[†]—OSR and AJCC stage not included in multivariate analyses to avoid complete exclusion of CRC/AAA patients respectively.

Increasing age is a risk factor for sarcopenia, though it is also recognised that skeletal muscle loss can occur in younger populations [8, 30]. There appears to be an association between increasing age and elevated magnitude of the systemic inflammatory response, which is a potential underlying mechanism explaining loss of skeletal muscle with age [31]. However, the adjusted analyses in the present study support a mechanistic role for elevated magnitude of the SIR independent of age. Moreover, both age and SIG are independently associated with both low SMI and low SMD, irrespective of disease state.

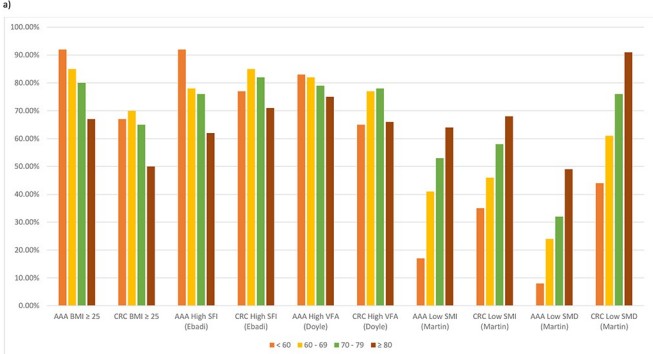
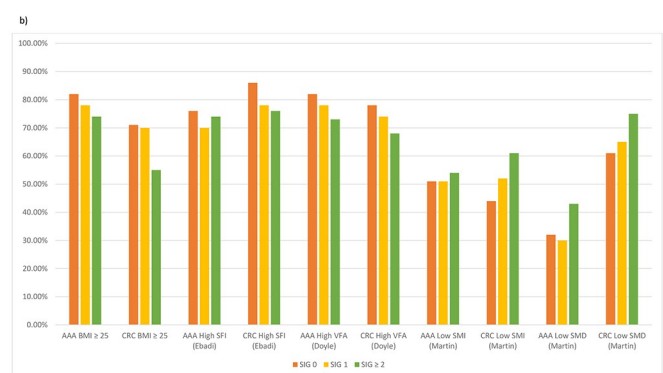

**Fig 1. The proportion of patients with abnormal BMI and CT-BC parameters grouped by a) age and b) SIG in patients with AAA and CRC, n = 1695.**
AAA: abdominal aortic aneurysm. CRC: colorectal cancer. BMI: body mass index. SIG: systemic inflammatory grade. SFI: subcutaneous fat index. VFA: visceral fat area. SMI: skeletal muscle index. SMD: skeletal muscle density.

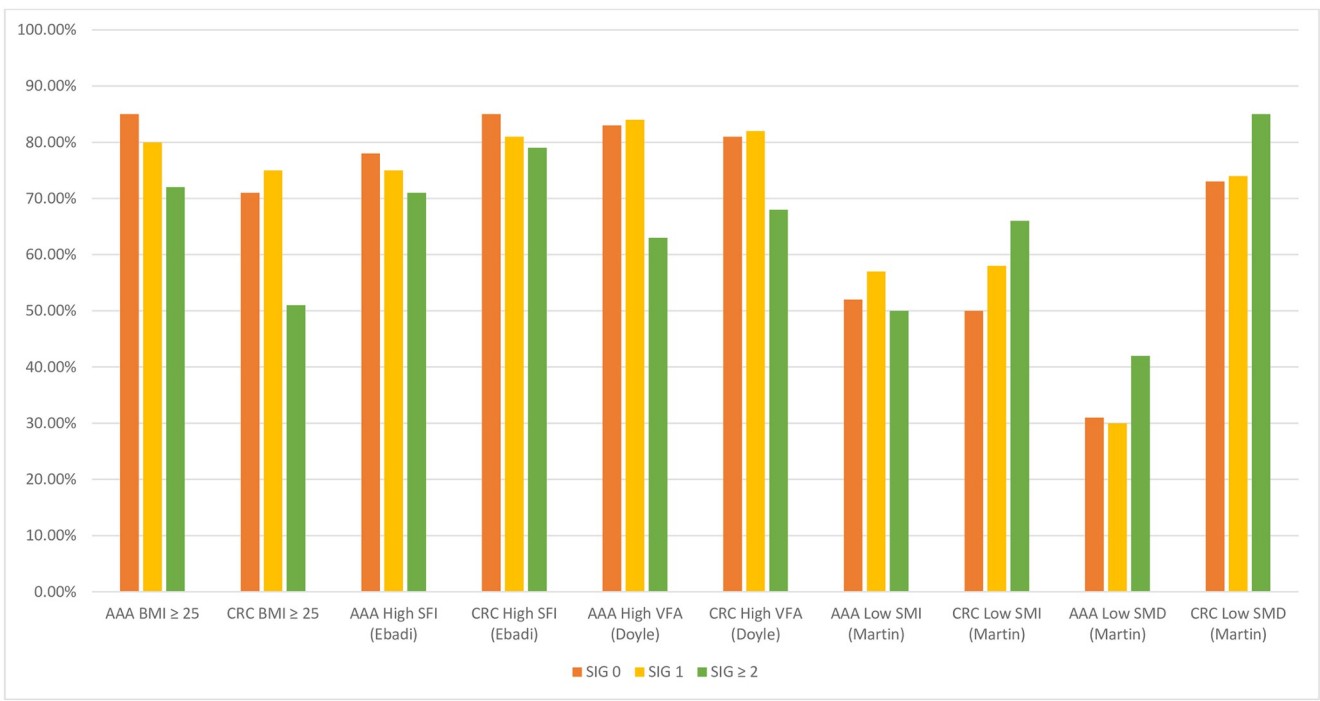

**Fig 2. The proportion of patients with abnormal BMI and CT-BC parameters grouped by SIG in patients with AAA and CRC age 70–79, n = 672.**
AAA: abdominal aortic aneurysm. CRC: colorectal cancer. BMI: body mass index. SIG: systemic inflammatory grade. SFI: subcutaneous fat index. VFA: visceral fat area. SMI: skeletal muscle index. SMD: skeletal muscle density.

The basis of the relationship between SMD and systemic inflammation is incompletely understood. However, it may be that systemic inflammation primarily influences muscle quality rather than quantity. If this were to prove to be the case then it might explain the profound effect of the SIR on physical function in patients with cancer [32]. These results also have implications for our understanding of the aetiology and definition of cachexia syndromes. For example, if such results are confirmed across disease states it may that more priority, compared with SMI, should be given to SMD and the importance of fat infiltration into skeletal muscle in determining clinical outcome. Clearly, this would suggest that muscle density (quality) rather than muscle mass would become a more meaningful therapeutic targets to optimise prognosis in a range of conditions.

The relationship between inflammation and muscle density has been reported in non-cancer populations, and attempts to define the mechanistic pathways suggest that cytokines such as TNF, IL-6, and IL-1β may stimulate proteolysis and inhibit anabolism, leading to muscle breakdown and wasting [33]. IL-1β blockade has been shown to reduce cardiac morbidity in patients with established cardiovascular disease [34]. Further prospective interventional studies investigating these potential therapeutic targets are required. In particular, SMD measurement appears to be associated with intramuscular lipid content [11]. Furthermore, intramyocellular lipid content is increased in the presence of cancer, and increased further in patients with clinical features of the cancer cachexia syndrome [35]. White adipose tissue, the predominant form of mammalian adipose tissue, is known to play an active role in the inflammatory response, potentially through the expression of adipokines which act as pro-inflammatory mediators [36]. Additionally, an association between inflammation and ectopic fat deposition in both liver and skeletal muscle has been described [37]. To date, radiological

hepatic architecture in relation to inflammation has not been directly compared between non-cancer and cancer cohorts, and investigation of this potential association may allow for greater understanding of the differences in CT-BC.

Mitochondria are the predominant organelles found in skeletal muscle, and have a role in inflammatory signalling [38], with potential pathways via reactive-oxygen species and the NLRP3 inflammasome [39]. Mitochondrial loss "mitophagy" has been described in relation to aging and chronic illness [40, 41]. The response of mitochondria to systemic inflammation in patients with cancer is uncertain. Downregulation of mitochondria, and mitophagy with amino acid release in response to both age and tumour-mediated inflammation may result in mitochondrial loss with preservation of the sarcomere, and subsequent fat deposition may explain the preferential SMD loss compared to SMI loss in the CRC cohort.

### Limitations

CT-BC parameters may vary depending on the phase of CT which is analysed, though there is uncertainty around the effects of phase and contrast on recorded values [28]. It appears that SMD is the most susceptible CT-BC parameter to such potential confounding. In order to mitigate this, we repeated our analyses using transformed SMD values as per the formula proposed by Rollins *et al* [28], which did not alter results. Given the uncertainty around the true effect that phase has on SMD values, we chose to report the non-transformed values in the main analysis. Rollins et al report that, as compared to the portal venous phase, arterial phase CTs may underestimate SMD values [28]. Given the present study observed lower SMD in the CRC cohort (portal venous phase) as compared to the AAA cohort (arterial phase) it is therefore likely that the effect of phase is underestimating rather than overestimating the strength of our observations. However, the conflicting results from previously published small series warrant further investigation to better understand the role of CT phase.

The potentially confounding effect of pharmacological therapies such as steroids, chemotherapy, statins on muscle tissue and the systemic inflammatory response may be considered. These data were not available for the present study cohorts, and is therefore a potential source of bias. However, typical UK practice does not advocate neoadjuvant steroid or chemotherapy for colonic cancer, therefore any effect is likely to be small. The association between statins and the results of interest in the present study would be an important area for further research. Nevertheless, it is clear from the present work that aging and systemic inflammation are important determinants of loss of muscle mass and content, independent of disease setting.

In the present study nutritional status may have been a potential confounding factor, however, the patient groups studied (i.e. patients undergoing intervention for AAA and patients undergoing surgery for colorectal cancer) are not recognised to be at high nutritional risk. For example, Chaar and colleagues reported that in patients undergoing intervention for AAA the mean (SE) pre-operative Nutritonal Risk Index was 103.6 (1.4); with values less than 100 conferring malnutrition [42]. Similarly, in a cohort of 363 patients undergoing surgery for CRC, 79% were classed as low risk of malnutrition based on MUST score [43]. Additional limitations include retrospective study design and missing data for some participants, however the present study is the largest comparative study of CT-BC in non-cancer and cancer patients to date.

### Conclusions

Low SMI was prevalent and similar non-cancer and cancer cohorts (AAA and CRC). Low SMD and systemic inflammation were more prevalent in patients with CRC than with AAA. Increasing age and systemic inflammation appear to be important determinants of loss of skeletal muscle mass and quality.

## Supporting information

**S1 Fig. Participant flow chart and inclusion into the study.** AAA: Abdominal aortic aneurysm. CRC: Colorectal cancer. WoSCAN: West of Scotland Cancer Network.
(TIF)

**S1 Table. The calculation of inflammation-based prognostic scores using pre-operative blood results.**
(DOCX)

**S2 Table. Classification of abnormal CT-derived body composition.**
(DOCX)

## Acknowledgments

The author would like to acknowledge the contributions of the Academic Department of Surgery, University of Glasgow, and the Departments of Vascular Surgery at NHS Tayside, NHS Grampian, and NHS Lanarkshire.

## Author Contributions

**Conceptualization:** Nicholas A. Bradley, Ross D. Dolan, Allan M. Golder, Campbell S. D. Roxburgh, Graeme J. K. Guthrie, Donald C. McMillan.

**Data curation:** Nicholas A. Bradley, Josh McGovern, Allan M. Golder.

**Formal analysis:** Nicholas A. Bradley, Josh McGovern.

**Methodology:** Nicholas A. Bradley, Ross D. Dolan, Allan M. Golder, Graeme J. K. Guthrie, Donald C. McMillan.

**Project administration:** Campbell S. D. Roxburgh, Graeme J. K. Guthrie.

**Resources:** Graeme J. K. Guthrie, Donald C. McMillan.

**Software:** Ross D. Dolan, Allan M. Golder.

**Supervision:** Campbell S. D. Roxburgh, Graeme J. K. Guthrie, Donald C. McMillan.

**Writing – original draft:** Nicholas A. Bradley, Josh McGovern, Donald C. McMillan.

**Writing – review & editing:** Nicholas A. Bradley, Josh McGovern, Ross D. Dolan, Allan M. Golder, Campbell S. D. Roxburgh, Graeme J. K. Guthrie, Donald C. McMillan.

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
