## [Decision Letter · Decision Letter 0]

3 Jan 2024

PONE-D-23-29602CT-derived body composition: differential association with disease, age and inflammation in a retrospective cohort study.PLOS ONE

Dear Dr. Bradley,

Thank you for submitting your manuscript to PLOS ONE. After careful consideration, we feel that it has merit but does not fully meet PLOS ONE’s publication criteria as it currently stands. Therefore, we invite you to submit a revised version of the manuscript that addresses the points raised during the review process.

We look forward to receiving your revised manuscript.

Kind regards,

Vincenzo Lionetti, M.D., PhD

Academic Editor

PLOS ONE

Journal Requirements:

2. In this instance it seems there may be acceptable restrictions in place that prevent the public sharing of your minimal data. However, in line with our goal of ensuring long-term data availability to all interested researchers, PLOS’ Data Policy states that authors cannot be the sole named individuals responsible for ensuring data access (http://journals.plos.org/plosone/s/data-availability#loc-acceptable-data-sharing-methods).

4. We note that there is identifying data in the Supporting Information file <Supplemental Table 2.docx>. Due to the inclusion of these potentially identifying data, we have removed this file from your file inventory. Prior to sharing human research participant data, authors should consult with an ethics committee to ensure data are shared in accordance with participant consent and all applicable local laws.

-Location data

Please remove or anonymize all personal information (name), ensure that the data shared are in accordance with participant consent, and re-upload a fully anonymized data set. Please note that spreadsheet columns with personal information must be removed and not hidden as all hidden columns will appear in the published file.

**Additional Editor Comments:**

All issues raised by expert reviewers ae required.

Reviewers' comments:

Reviewer's Responses to Questions

**Comments to the Author**

1. Is the manuscript technically sound, and do the data support the conclusions?

Reviewer #1: Yes

Reviewer #2: Yes

2. Has the statistical analysis been performed appropriately and rigorously? 

Reviewer #1: Yes

Reviewer #2: Yes

3. Have the authors made all data underlying the findings in their manuscript fully available?

Reviewer #1: No

Reviewer #2: Yes

4. Is the manuscript presented in an intelligible fashion and written in standard English?

Reviewer #1: Yes

Reviewer #2: Yes

5. Review Comments to the Author

Reviewer #1: In this interesting paper, the Authors compare skeletal muscle quantity and quality reduction in AAA and CRC patients, and evaluate if systemic inflammation influence SMI and SMD reduction. The work is correctly conducted, the rationale is clear and the paper clearly written.

I have only some minor suggestions:

-introduction: "Low SMD appears to be associated with intramuscular lipid content[11], and may represent poor quality muscle through replacement of healthy muscle tissue with adipose tissue.". I think that SMD represents fat inside the muscle fibers (while intermuscolar adipose tissue area or IMAT represents adipose tissue infiltrating the muscle, so within fibers and beyond the fascia), so probably low SMD does not exactly represent replacement of muscle with adipose tissue, but instead the accumulation of fat droplets within muscle cells. Of course the poor effect on muscle quality is the same.

Methods: it's not clear how the SMD values were made homogeneous between the two cohorts. "in order to mitigate the potential risk of bias in SMD readings between these 2 phases SMD analyses were repeated using values in the AAA cohort transformed as per Rollins et al". First, Rollins et al analyzed a small cohort and even if here the Authors use the equations proposed by Rollins et al, the use of two different phases is still a potential source of bias. Second, how was the equation used? Did the Authors obtain portal venous phase SMD values also on the AAA cohort by using the equation? or they obtain unenhanced SMD for the whole cohort? If the cut-off used for low SMD definition is that from Martin et al., it was obtained on unenhanced CT, hence defining low SMD on enhanced CT by using Martin's cut-off can introduce misclassification. Maybe using SMD as a continuous variable and reporting also mean and SD in different groups can be useful?

Reviewer #2: I read with extreme interest the article by Bradley et al. on CT-derived body composition and its differential association with disease, age, and inflammation. Although this is a retrospective cohort study, it sheds attractive light on prognosis and body habitus in two chronic disease settings, the oncologic (colon) and the vascular (aortic aneurysm) settings. The message is that aging and systemic inflammation are ubiquitous determinants of loss of muscle mass and content, independently of disease setting. The paper is well written. I only have some concerns inherent in the possible association with broad-spectrum anti-inflammatory pharmacological therapies (e.g., in cancer, steroids, aortic aneurysms, statins). Do the authors have bioumoral data such as troponin or CPK available? These indicators are associated with chronic vascular and cancer damage and could shed additional helpful light in the context. How was the possible difference in dietary habits between the two populations considered?

6. PLOS authors have the option to publish the peer review history of their article (what does this mean?). If published, this will include your full peer review and any attached files.

Reviewer #1: No

Reviewer #2: No

---

## [Decision Letter · Decision Letter 1]

21 Feb 2024

CT-derived body composition: differential association with disease, age and inflammation in a retrospective cohort study.

PONE-D-23-29602R1

Dear Dr. Bradley,

We’re pleased to inform you that your manuscript has been judged scientifically suitable for publication and will be formally accepted for publication once it meets all outstanding technical requirements.

Kind regards,

Vincenzo Lionetti, M.D., PhD

Academic Editor

PLOS ONE

Additional Editor Comments (optional):

Reviewers' comments:

Reviewer's Responses to Questions

**Comments to the Author**

1. If the authors have adequately addressed your comments raised in a previous round of review and you feel that this manuscript is now acceptable for publication, you may indicate that here to bypass the “Comments to the Author” section, enter your conflict of interest statement in the “Confidential to Editor” section, and submit your "Accept" recommendation.

Reviewer #2: All comments have been addressed

2. Is the manuscript technically sound, and do the data support the conclusions?

Reviewer #2: Yes

3. Has the statistical analysis been performed appropriately and rigorously? 

Reviewer #2: Yes

4. Have the authors made all data underlying the findings in their manuscript fully available?

Reviewer #2: Yes

5. Is the manuscript presented in an intelligible fashion and written in standard English?

Reviewer #2: Yes

6. Review Comments to the Author

Reviewer #2: Manuscript has been improved by the authors. All comments have been addressed. It is now worth publishing

7. PLOS authors have the option to publish the peer review history of their article (what does this mean?). If published, this will include your full peer review and any attached files.

Reviewer #2: No

---

## [Editor Report · Acceptance letter]

7 Mar 2024

PONE-D-23-29602R1 

PLOS ONE

Dear Dr. Bradley, 

I'm pleased to inform you that your manuscript has been deemed suitable for publication in PLOS ONE. Congratulations! Your manuscript is now being handed over to our production team.

Kind regards, 

on behalf of

Prof. Vincenzo Lionetti 

Academic Editor

PLOS ONE